# Environmental complexity is more important than mutation in driving the evolution of latent novel traits in *E. coli*

**Shraddha Karve** [1,2] ✉ **& Andreas Wagner** [1,3,4,5] ✉

Recent experiments show that adaptive Darwinian evolution in one environment can lead to the emergence of multiple new traits that provide no immediate benefit in this environment. Such latent non-adaptive traits, however, can become adaptive in future environments. We do not know whether mutation or environment-driven selection is more important for the emergence of such traits. To find out, we evolve multiple wild-type and mutator *E. coli* populations under two mutation rates in simple (single antibiotic) environments and in complex (multi-antibiotic) environments. We then assay the viability of evolved populations in dozens of new environments and show that all populations become viable in multiple new environments different from those they had evolved in. The number of these new environments increases with environmental complexity but not with the mutation rate. Genome sequencing demonstrates the reason: Different environments affect pleiotropic mutations differently. Our experiments show that the selection pressure provided by an environment can be more important for the evolution of novel traits than the mutational supply experienced by a wild-type and a mutator strain of *E. coli*.

Most experimental and theoretical work in evolutionary biology focuses on adaptive traits. However, in recent years experimental evidence has been mounting that such traits may be outnumbered by latent traits without immediate benefits, i.e., traits that are not immediately adaptive in their environment of origin, but that may become adaptive in the right kind of future environment[1–5]. Such potentially adaptive traits are important in evolution, because they can create new morphological structures or physiological abilities, and they can give rise to new ecological niches[6,7]. The existence of many such traits has been reported both in the wild[4,8] and in the laboratory[9,10]. For example, we recently evolved *E. coli* populations in environments harbouring single antibiotics and showed that these populations evolve the ability to survive in multiple novel environments that inhibit bacterial growth through mechanisms different from the antibiotic they evolved in ref.

5. We refer to the newly acquired (and non-adaptive) viability in any one such environment as a latent novel trait. Here we study such traits to address a long-standing debate on the relative importance of mutation and selection in evolutionary biology.

Ever since Darwin proposed his theory of evolution, biologists have debated the relative roles of mutation supply and natural selection in Darwinian evolution[11–14]. On the one hand, 'selectionists' such as Weismann, Wallace and Darwin himself asserted the dominant role of natural selection that is exerted by the environment[11,14]. On the other hand, 'mutationists' like Morgan and Bateson argued that variation provided by mutations acts as the creative force during evolution, and that natural selection is merely a sieve retaining favourable variation[13]. With every major advancement in evolutionary biology, like the rediscovery of Mendel's work or the development of population

[1]Department of Evolutionary Biology and Environmental Studies, University of Zurich, Zurich, Switzerland. [2]Ashoka University, NH 44, Rajiv Gandhi Education City, Sonipat 131029, India. [3]The Santa Fe Institute, Santa Fe, NM, USA. [4]Stellenbosch Institute for Advanced Study (STIAS), Wallenberg Research Centre at Stellenbosch University, Stellenbosch 7600, South Africa. [5]Swiss Institute of Bioinformatics, Quartier Sorge-Batiment Genopode, Lausanne, Switzerland. ✉e-mail: shraddha.karve@gmail.com; andreas.wagner@ieu.uzh.ch

genetics, the balance kept shifting in favour of either mutationists or selectionists[11,12]. The prevailing view today is that the selection exerted by the environment is the dominant force, rather than the supply of mutations, in shaping the evolution of adaptive traits. When populations face a new environment, existing alleles present as standing variation can sweep to fixation more rapidly and accelerate the speed of adaptation compared to alleles that are newly created by mutation[15,16]. Moreover, selection exerted by environment can modulate the mutation supply itself[17]. But this view of 'selection over mutations' has been tested only in the context of adaptive traits. It has not been extended to or contradicted by the evolution of non-adaptive novel traits. We focus on such traits for two reason. First, the forces that determine their evolution are more poorly understood than those of better-studied adaptive traits. Second, because they are not directly subject to selection in their environment of origin, they might be good candidates to provide evidence for the long-neglected role of mutation supply.

On the one hand, one may argue that mutations should be the driving force behind the origin of latent traits. Because such traits are neither beneficial nor deleterious in the environment in which they originate, natural selection in this environment should not affect them. If so, their incidence should be primarily determined by the rate at which mutations bring forth new genetic variation, i.e., by the mutation supply. On the other hand, while natural selection acts on adaptive traits, it may indirectly act on the genetic variation that affects non-adaptive traits as well. The environment in which evolution occurs can thus modulate the amount and nature of genetic variation that is retained during evolution.

To distinguish these possibilities, we here ask whether the amount of genetic variation supplied by mutation or the selection pressure imposed by an environment is more important for the evolution of latent novel traits. To answer this question, we evolve replicate *E. coli* populations at two different mutation rates, and in both simple environments that contain a single antibiotic, and in complex environments that contain multiple antibiotics. We refer to these environments collectively as evolution environments. Specifically, we evolve a wild-type and a mutator strain with a twenty-two-fold higher mutation rate than the wild-type[18]. We evolve the mutator strain in the same five simple environments as in a previously reported experiment, in which we had evolved the wild-type strain in five environments containing one antibiotic each[5]. In addition, we evolve both the wild-type and the mutator strain in three complex environments with either three or five antibiotics. After experimental evolution we determine whether the evolved populations have become viable in dozens of phenotyping environments that are different from the evolution environments of our experiments, and that the ancestral populations are not viable in. We find that for the wild-type and mutator strains the number of latent novel traits that evolve depend on the complexity of the environment but not on the mutation supply. Genome sequencing suggests that pleiotropic mutations in multi-drug resistance genes, such as the AcrAB-TolC efflux system, are important for the emergence of these novel traits. In sum, at the mutation rates of our wild-type and mutator strains, selection exerted by the environment is the key force in the evolution of novel traits without immediate benefits.

## Results

### The mutation rate does not limit the emergence of novel traits without immediate benefit

To study how an increased supply of mutations might affect the evolution of viability in new environments, we used both a wild-type strain of *E. coli* and a 'mutator' strain with a twenty-two-fold higher mutation rate (Methods)[18]. Prior to experimental evolution we determined the phenotypic differences between the wild-type and mutator strains. Such differences may be generated by the mutator strain's intrinsically higher mutation rate, which may cause more mutations and their ensuing phenotypic effects even during initial strain cultivation[18]. To identify phenotypic differences between the two strains, we used a set of ten Biolog Phenotyping microarrays (PM11-20, Biolog Inc., USA). These microarrays comprise 236 different environments that inhibit microbial growth. Each environment harbours a different antimicrobial agent chosen from a wide range of categories, including antibiotics, detergents, surfactants, ion chelators, oxidising agents, and pyridine analogues. To determine a strain's viability in each of these phenotyping environments, we randomly chose two clones of the strain from an LB (Luria Bertani medium) agar plate incubated overnight, and required that at least one of the clones was able to survive and grow in the environment ($OD_{600}$ after 48 h of growth >0.3, Methods).

By this criterion, our wild-type strain was inviable in 95 of the 236 environments, as reported previously[5]. The mutator strain was inviable in 58 of the 236 environments (Fig. 1A). The mutator strain was thus viable in more (178 = 236−58) environments than the wild-type strain (141 = 236−95). Both strains were inviable in the same 52 environments. In other words, the wild-type strain was inviable in 43 (=95−52) environments where the mutator strain was viable, whereas the mutator strain was inviable in only 6 (=58−52) environments where the wild-type strain was inviable (Fig. 1A). The environments in which the wild-type or the mutator strains were viable harbour antimicrobials with diverse mechanisms of action. They include iron chelators like 2,2-dipyridyl[19] and lawsone[20], the oxidising agent diamide[21–23], as well as several antibiotics like ciprofloxacin, polymyxin B, and vancomycin (Table S1). We sequenced the genomes of both mutator clones using Illumina HiSeq (Illumina, CA, USA) to at least 30-fold genomic coverage per clone to identify candidate mutations that may have increased the mutator's viability, and found five such mutations (Supplementary note S1, Table S2 for the details).

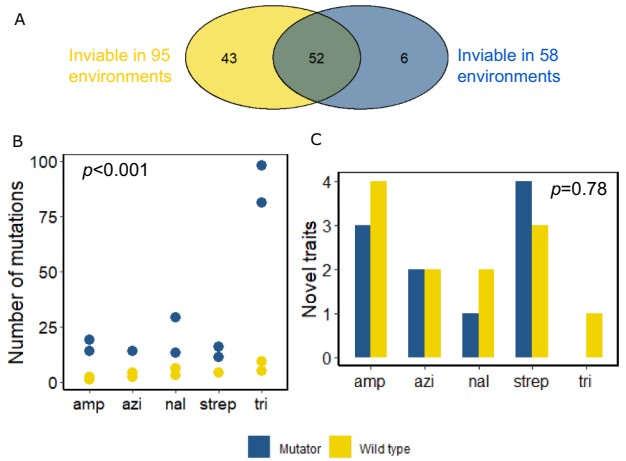

**Fig. 1 | Mutation supply does not limit the emergence of novel traits without immediate benefit. A** Before conducting any evolution experiments, we determined the viability of both our ancestral wild-type and ancestral mutator strain in 236 different environments with Biolog phenotyping microarrays. The mutator strain was inviable in 58 environments, whereas the wild-type was inviable in 95 environments, as previously reported[5]. In 52 environments both the wild-type and mutator ancestors were inviable. **B** After ~200 generations of experimental evolution in five different simple (single-antibiotic) environments (*x*-axis) the number of mutations (*y*-axis) in evolved mutator clones (blue circles) was significantly higher than in the evolved wild-type clones (yellow circles) (Two-way ANOVA, F = 115.77, df = 1, $p = 8 \times 10^{-7}$). **C** We determined the number of phenotyping environments in which an evolved strain had acquired viability (*y*-axis) out of the total number of 52 environments in which both the ancestors had not been viable before evolution. This number is statistically indistinguishable for the evolved wild-type (yellow bars) and the evolved mutator clones (blue bars) in each of the five simple environments (*x*-axis, Two-sided Wilcoxon rank sum test, W = 14.5, *n* = 5 and 5, *p* = 0.78). Source data are provided as a Source Data file.

Starting from the ancestral mutator strain, we next conducted five separate evolution experiments identical to those we previously described for the wild-type strain[5]. Specifically, we performed each of these experiments in five different environments, where each environment contained one of five different single antibiotics. We refer to these environments as simple evolution environments. The antibiotics are ampicillin (amp), azithromycin (azi), nalidixic acid (nal), streptomycin (strep), and trimethoprim (tri). We choose these antibiotics because they have distinct cellular targets and different modes of action[24,25].

Analogous to our previously described experiments with the wild-type strain, we evolved eight replicate populations of the mutator strain for ~100–200 generations in each of the five simple environments (Methods, Table S3), until all the populations could grow at the $IC_{90}$ of the antibiotic they evolved in. The $IC_{90}$ is the concentration of an antibiotic that kills 90% of all cells in the wild-type ancestral strain[1]. At the end of the evolution experiment, we identified two representative evolved clones from each antibiotic environment for further analyses. We chose these clones to represent the central tendency of the growth rates of the evolved populations (Methods, Table S4 and Fig. S2). We then sequenced all evolved clones using Illumina HiSeq to at least 30-fold coverage (Illumina, CA, USA, Methods), which also confirmed that all evolved mutator clones retained the 103 bp insertion upstream of the *mutL* gene that endows them with their higher mutation rate.

As expected, we observed significantly more genomic mutations in the evolved mutator clones than in the evolved wild-type clones (Fig. 1B, Two-way ANOVA, $F = 115.77$, df = 1, $p = 8 \times 10^{-7}$, Methods). This difference ranged between a threefold greater number of mutations for the mutator in the streptomycin environment, and a twelve-fold greater number in the trimethoprim environment. Thus, the evolved mutator clones did not only experience more mutations as a result of their higher mutation rate, they also retained more mutations after experimental evolution.

We next asked whether our evolved strains had become viable in any of the 236 phenotyping environments of the Biolog phenotyping microarrays. We called viability novel in a given environment if both clones that had evolved on the same antibiotic were able to survive and grow in this environment, even though neither ancestral strains were able to. For instance, both the wild-type and mutator ancestral clones could not grow on the antibiotic spectinomycin which inhibits protein synthesis[26], but both mutator clones evolved in ampicillin could. This novel ability was without any immediate benefit, because spectinomycin was not present in the ampicillin environment in which the clones evolved. In four out of five simple environments, except the evolution environment with streptomycin, wild-type clones had evolved more novel traits than mutator clones. However, this difference was not statistically significant (Two-sided Wilcoxon rank sum test, $W = 14.5$, $n = 5$ and 5, $p = 0.78$). The antimicrobials on which these clones had evolved viability inhibit growth through diverse mechanisms but these mechanisms differed from that of the antibiotic in the respective evolution environment in many phenotyping environments (Supplementary table 10.)

More importantly, however, evolved mutator clones did not show more novel traits (Fig. 1C), even though they experienced more mutations and retained more of the resulting genetic variation during evolution (Fig. 1B). Thus, the supply of mutations does not limit the evolution of latent novel traits at mutation rates that exceed those of the wild-type.

## Complex antibiotic environments facilitate the emergence of latent novel traits

We next investigated the role of the selection environment in the evolution of novel traits without immediate benefit. So far we had

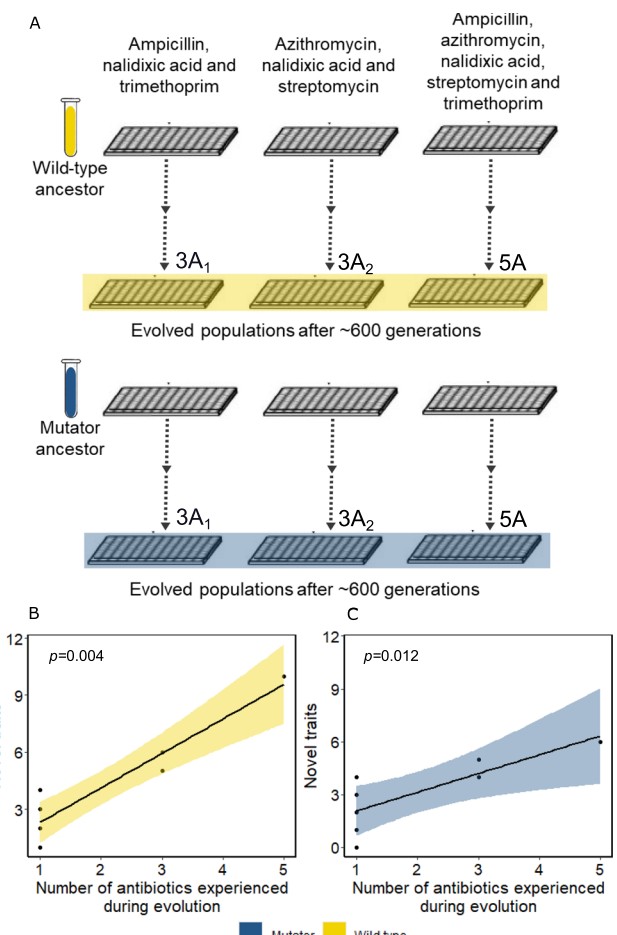

**Fig. 2 | Evolution of novel traits without immediate benefit in complex antibiotic environments A.** Experimental evolution design in the three different complex environments $3A_1$, $3A_2$ and 5A. We evolved eight populations of wild-type and mutator strains (forty-eight populations in total, Methods) in increasing concentrations of the indicated antibiotics for ~600 generations. At the end of the evolution experiment all populations could grow in the environment that contained all three ($3A_1$ and $3A_2$) or five (5A) antibiotics at their respective $IC_{90}$. **B, C** The number of novel traits is significantly and positively correlated with the number of antibiotics experienced during evolution (x-axis) for both the wild-type (**B**, Spearman's correlation, $n = 8$, $R = 0.87$, $p = 0.004$) and the mutator strain (**C**, Spearman's correlation, $n = 8$, $R = 0.82$, $p = 0.012$). The shaded region represents the 95% confidence intervals. Source data are provided as a Source Data file.

studied simple environments that harboured only a single antibiotic. We next evolved our strains in three complex environments which contained more than one antibiotic (Fig. 2A). Two out of the three environments contained three antibiotics each. Specifically, environment $3A_1$ (for 'three antibiotics') harboured ampicillin, nalidixic acid, and trimethoprim. Environment $3A_2$ harboured azithromycin, nalidixic acid, and streptomycin. The remaining complex environment '5A' contained all five antibiotics that we had used in the evolution experiment with simple environments (Fig. 1C). We performed six evolution experiments in the complex environments, three for the wild-type strain and three for the mutator strain. In each experiment we evolved eight replicate populations of wild-type or mutator *E. coli* until the populations could grow at the $IC_{90}$ of each of the antibiotics present in the environment. We used a procedure which ensured that the populations evolved for a similar amount of time to acquire this ability (~600 generations or ~147 days, Methods, Supplementary note S2). After experimental evolution, we chose two representative evolved wild-type and mutator clones from each of

the three environments for phenotyping and genome sequencing (2 clones × 2 strains × 3 environments = 12 clones in total, Methods, Table S5 and Fig. S4).

The greater the number of antibiotics present in the evolution environment was, the greater was the percentage of phenotypic environments in which our strains gained viability during experimental evolution. This holds both for the wild-type strain (Fig. 2B, Spearman's correlation, $n = 8$, $R = 0.87$, $p = 0.004$, Methods) and for the mutator strain (Fig. 2C, Spearman's correlation, $n = 8$, $R = 0.82$, $p = 0.012$, Methods).

In sum, increasing the complexity of the environment in which our strains evolved increases the number of novel traits without immediate benefit, irrespective of the mutation supply. Thus, the selection environment plays the predominant role in the evolution of novel traits without immediate benefits.

## The nature of genetic variation determines the extent of novel trait evolution

An important confounding factor in our analysis is the time that our populations spent evolving. Specifically, evolution in complex environments lasted for almost ~600 generations, approximately three times longer than evolution in simple environments (~100–200 generations). As a result, wild-type and mutator populations that evolved in complex environments have experienced more mutations than their counterparts that evolved in simple environments. In consequence, the higher incidence of novel trait evolution in complex environments (Fig. 2B, C) could be caused by this higher number of mutations, as a result of the longer time our populations spent evolving in complex environments. In fact, not only the supply of mutations but also the number of mutations retained after experimental evolution is significantly higher in complex environments for both the wild-type and mutator strain (Fig. S5).

To control for this confounding factor, we first quantified the partial correlation between environmental complexity and the number of evolved novel traits, while controlling for the number of generations. In this analysis, high environmental complexity remained significantly associated with a high number of evolved novel traits, both for the wild-type (partial Spearman's $R = 0.95$, $n = 8$, $p = 0.0007$) and the mutator strain (partial Spearman's $R = 0.95$, $n = 8$, $p = 0.0009$).

Second, we compared the extent of novel trait evolution for clones evolved in the two kinds of complex environments (3A and 5A), because our populations had evolved for an identical amount of time in these environments. This analysis suggests a predominant role of environmental complexity, but not for the number of mutations, in driving novel trait evolution (Fig. S6). However, it also lacks statistical power, because we evolved populations only in a single environment containing five antibiotics.

Third, we compared mutator clones evolved in simple antibiotic environments to wild-type clones evolved in complex antibiotic environments. A simple calculation shows that these two kinds of clones experienced a similar number of mutations during experimental evolution (Supplementary note S3). In other words, the increased time spent by wild-type clones in complex antibiotic evolution environment compensated partly for the higher mutation rate of mutator clones. In addition, our genomic analysis showed that the number of genetic variants retained after evolution did not differ significantly between the mutator clones evolved in the simple environments and wild-type clones evolved in the complex environments (Fig. 3A, Two-sided Wilcoxon rank sum test, $n = 10$ and 6, $W = 28$, $p = 0.86$). Based on these observations, we reasoned that it may be appropriate to compare novel trait evolution between these two types of clones. This comparison also supports our previous observations. That is, wild-type clones from complex environments evolved a significantly higher number of novel traits without immediate benefit than mutator clones evolved in simple environments (Fig. 3B, Two-

sided Wilcoxon rank sum test, $n = 5$ and 3, $W = 0$, $p = 0.035$). Once again, selection exerted by the evolution environment, and not the mutation supply or the amount of genetic variation retained during evolution, is the key force behind the evolution of novel traits without immediate benefits.

## Complex environments preferentially select for pleiotropic mutations that can increase viability in multiple environments

We next asked what kinds of genetic variation may be responsible for the higher prevalence of novel traits in complex environments. To answer this question, we compared again the mutator clones evolved in simple environments and the wild-type clones evolved in complex environments, because they received a similar mutation supply, and harboured similar amounts of genetic variation.

Specifically, we first focused on genetic variants in genes that encode cellular targets of antibiotics in the evolution environment, or proteins that directly interact with such targets. Mutations in such genes do not only cause resistance to these antibiotics, but have pleiotropic effects that can bring forth novel traits[1,5,27,28]. We found pertinent variants in all evolved clones, except for one mutator clone evolved in azithromycin (Fig. 3C, Table S6). However, clones evolved in complex environments harboured many more mutations in antibiotic target genes than clones evolved in simple environments. Specifically, six evolved clones from complex environments harboured 24 variants in genes encoding antibiotic targets, whereas the greater number (ten) of mutator clones evolved in simple environments harboured merely 16 variants in such genes. Many of these variants have known pleiotropic effects. For example, the *gyrA* gene was mutated at least once in all the wild-type clones evolved in complex environments, but only in two of the mutator clones evolved in a simple antibiotic environment, namely that harbouring nalidixic acid. Mutations in *gyrA* can confer not just resistance against nalidixic acid, but also against β-lactams and aminoglycosides, likely by modifying the supercoiling of DNA and global gene expression with it[28,29]. Similarly, one of the wild-type clones evolved in the 3A$_2$ environment, which contained streptomycin, harboured a mutation in the gene *infB* (Fig. 3C), whereas none of the two mutator clones evolved on streptomycin harboured a mutation in *infB*. The gene encodes the translation initiation factor IF-2, which interacts closely with the 30S ribosomal subunit, the cellular target of streptomycin[30,31]. Mutations in *infB* can also confer resistance against macrolide antibiotics that target the 50S ribosomal subunit in protein synthesis[32]. In a similar vein, one of the wild-type clones evolved in the 3A$_1$ environment, but none of the mutator clones evolved on a single antibiotic, harboured a mutation in the gene *ampC*. Mutations in this gene can confer resistance to carbapenems and the combination drugs ceftolozane-tazobactam and ceftazidime-avibactam[33,34]. Likewise the gene *ftsI* was only mutated in one of the wild-type clones from 3A$_1$ environment (Fig. 3C). The genes codes for peptidoglycan D,D-transpeptidase[35], and mutations in it can also increase resistance to mecillinam, cephalexin, and sefsulodin[36]. Taken together, these observations suggest that complex environments select for the spreading of pleiotropic mutations that can affect viability in multiple environments.

Further support for this hypothesis comes from a closer examination of genes that encode proteins involved in multi-drug efflux, which are well-known to have pleiotropic effects in multiple environments[37–40]. Overall, we found twenty-nine mutations across ten different genes implicated in multi-drug resistance. Twenty-five of these mutations occurred in the wild-type clones evolved in complex environments, while only three occurred in mutator clones evolved in the simple environments (Fig. 3D, Table S6). Affected genes included those encoding global transcription regulators, such as emrR (mprA)[37,41], efflux pumps, such as mdt[38] and yojI[40,42], as well sensory and regulatory proteins that respond to environmental stress, such as envZ[39,43] and phoQ[28]. Mutations in *emrR* can increase resistance to

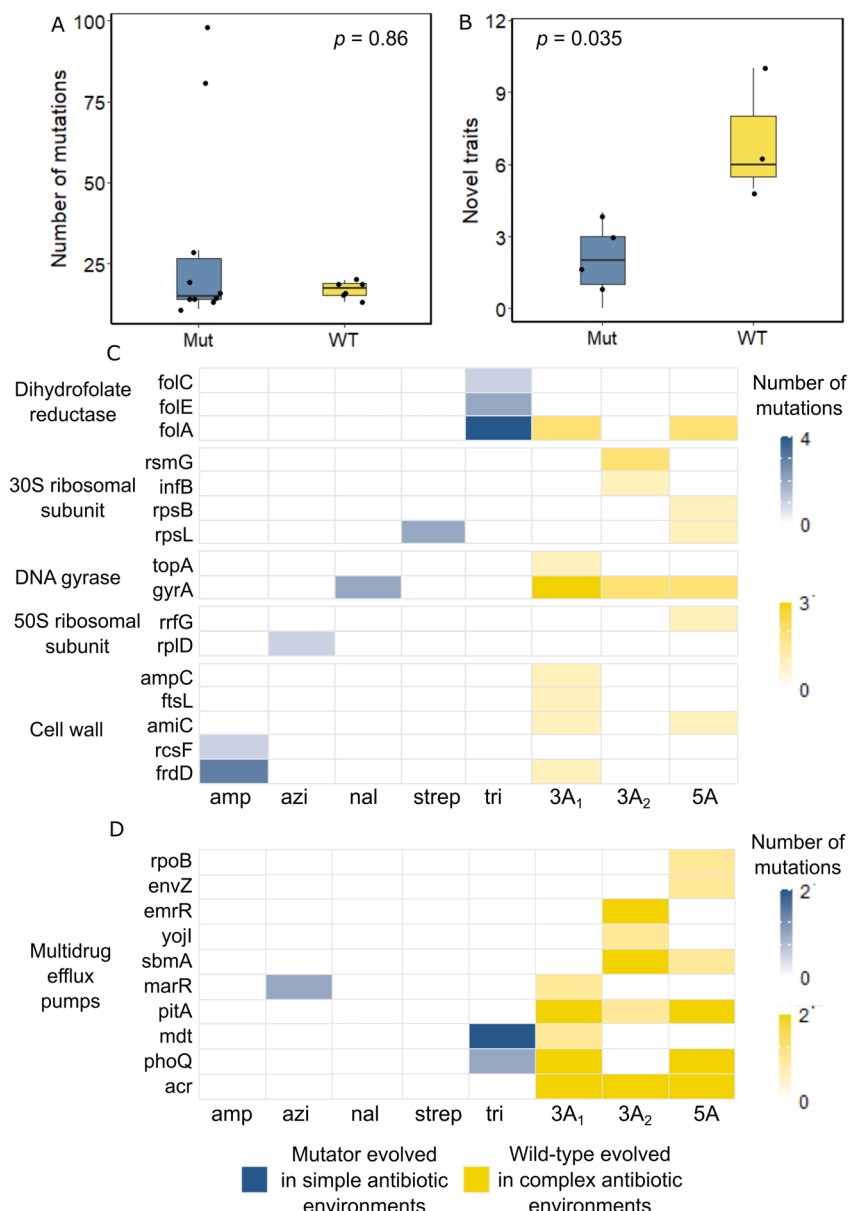

**Fig. 3 | The nature of retained genetic variation differs between populations evolved in simple and complex environments A.** The total number of retained genomic variants was statistically indistinguishable (Two-sided Wilcoxon rank sum test, $n = 10$ and 6, $W = 28$, $p = 0.86$) between mutator clones evolved in simple antibiotic environments (blue) containing a single antibiotic, and wild-type clones evolved in complex antibiotic environments (yellow) containing three (environments $3A_1$ and $3A_2$) or five (environment $5A$) antibiotics. **B** The number of evolved novel traits was significantly higher (Two-sided Wilcoxon rank sum test, $n = 5$ and 3, $W = 0$, $p = 0.035$) in wild-type clones evolved in complex antibiotic environments than in mutator clones evolved in simple antibiotic environments. In the (**A, B**), the boxes represent interquartile range while the solid line represents the median. The whiskers represent 1.5 times of the interquartile range. The circles located above the top whisker are outliers whose values are higher than 1.5 times the interquartile range (third quartile – first quartile) above the first quartile. **C** Number of mutations in the genes (second column from the left) that encode the cellular target (first column from the left) of proteins that directly interact with the cellular target of the antibiotic(s) experienced during experimental evolution, for mutator clones evolved in simple antibiotic environments (blue) and wild-type clones evolved in complex antibiotic environments (yellow). **D** Number of mutations in the genes (second column from the left) that are involved in multi-drug resistance for mutator clones evolved in simple antibiotic environments (blue) and wild-type clones evolved in complex antibiotic environments (yellow). For (**C, D**), each tile represents the total number of mutations we observed in a specific gene (second column from the left) for the two clones that had experienced a given antibiotic environment (bottom row) during experimental evolution. Source data are provided as a Source Data file.

many antibiotics, such as nalidixic acid, nitrofurantoin and erythromycin, while mutations in the gene *phoQ* can also confer increased resistance against antibiotics with diverse cellular targets, such as ampicillin, ciprofloxacin, nalidixic acid, kanamycin and tobramycin[28]. Similarly, mutations in the gene *envZ* can increase resistance to ampicillin, cefoxitin, ciprofloxacin, chloramphenicol, erythromycin, and tetracycline[28]. Most notably, all wild-type clones evolved in complex environments harboured at least one mutation in the genes

coding for the AcrAB-TolC efflux pump, while none of the mutator clones evolved in simple antibiotic environments did. AcrAB-TolC is an important multi-drug efflux system in *E. coli* that exports antibiotics with diverse cellular targets, as well as non-antibiotic toxins[44,45].

In sum, these observations help explain why complex antibiotic environments promote the evolution of latent traits without immediate benefit. They promote the spreading of pleiotropic mutations that can help bacteria become resistant against multiple antibiotics they

encounter during evolution. As a by-product, these mutations also convey viability in other environments that the bacteria have not encountered.

## Discussion

To our knowledge, this is the first study that experimentally addresses the relative importance of mutation and selection in the context of novel traits without immediate benefits. It can be difficult to identify such traits, because they are only potentially adaptive, and it can be even more difficult to study their evolution. We find that for the evolution of such latent novel traits, the selection pressure exerted by the environment in which populations evolve is more important than the amount of genetic variation provided by mutations. Increasing the complexity of the environment from one to three to five antibiotics results in the evolution of viability in multiple additional environments, whereas increasing the supply of mutations does not, despite an increase in the amount of genetic variation retained. This higher prevalence of novel traits in complex environments occurs at both low (wild-type) and high (mutator) mutation rates.

To vary the mutation supply we used ancestors with two different mutation rates, a wild-type strain and a mutator strain with a twenty-two fold higher mutation rate. We then evolved these strains under identical experimental conditions. Previous studies have shown that the frequency of bacterial mutator strains in natural or clinical environments is often higher than expected by mutation-selection balance[46–48]. High mutation rates can increase the rate of adaptation to drugs both in the laboratory and in the wild[49]. They may also lead to a greater number of latent novel traits during very brief periods of evolution. For example, the initial cultivation of our mutator strain before our evolution experiment resulted in a mutator ancestor with ~25 genomic mutations (Supplementary note S1), and an ability to grow in more phenotyping environments than our wild-type ancestor (Fig. 1A). In contrast, on the longer time scale of our evolution experiment, it is the wild-type and not the mutator that brings forth more novel traits in complex environments. This observation underscores that selection is the prevalent force behind the emergence of novel latent traits in our experiments.

The relative importance of selection and mutation may depend on the mutation supply itself. On the one hand, in strains with a mutation rate lower than the wild-type, the mutation supply may limit the origin of novel traits. On the other hand, at the higher than wild-type mutation rates of our experiments, multiple clones with equally beneficial variants may coexist in the same population. This can result in 'clonal interference', a process that reduces the efficacy of selection and can also limit the emergence of novel traits[50,51]. Our observations show that at wild-type mutation rates, the mutation supply does not limit the origin of novel traits. Had we compared our wild-type strain to an antimutator strain with a lower mutation rate[52], we might have found that fewer novel traits evolve in the anti-mutator. However, we deliberately did not use such a strain, because mutation rates far below our wild-type are rarely observed in nature[52]. To study how readily latent novel traits may evolve at mutation rates lower than that of the wild-type remains an exciting direction for future work. In addition, one could also replicate our experimental design in a bacterial species with a lower wild-type mutation rate than that of E. coli. Results of such studies, combined with our observations, can uncover the relationship between mutation rates and the emergence of latent novel traits over a wider range of mutation supplies.

Our genome analysis reveals that pleiotropic mutations are key for the evolution of novel traits, an observation that is consistent with previous studies. For instance, in a laboratory evolution experiment, E. coli populations became resistant to multiple antibiotics by acquiring mutations in transcriptional repressors of the antibiotic stress response, such as marR and mprA[28]. Similarly, beneficial mutations that increase the fitness of E. coli populations evolving in a glucose-limiting

environment can also improve fitness on carbon sources different from glucose[53]. Another striking example comes from E. coli 'deep-rough' mutants. They harbour mutations in the rfa operon, which help produce truncated lipopolysaccharides with pleiotropic effects on diverse traits, such as susceptibility to bacteriophages, antibiotics, and antimicrobial peptides[54]. Our previous work using a wild-type E. coli strain evolving in single antibiotic environments also highlights the importance of pleiotropy in the evolution of latent novel traits[5].

In the same vein, this study shows that the ability to produce many novel traits is linked to pleiotropic mutations that occur in genes required to combat multiple environmental stresses (Fig. 3D). More specifically, exposure to multiple antibiotics with diverse modes of action and cellular targets leads to the evolution of resistance through more than one mechanism. For instance, one wild-type clone that evolved in the complex 5A environment harboured mutations in the genes encoding cellular targets of four of the five antibiotics in this environment, as well as four mutations in genes involved in multi-drug resistance (Table S13, wild-type 5A(I)). In contrast, clones evolved in simple antibiotic environments usually harboured mutations in the genes encoding the cellular target of the respective antibiotic, but rarely showed multi-drug resistance mutations (Fig. 3C). A striking example of this contrast between simple and complex environments involves the genes encoding or regulating the AcrAB-TolC efflux pump. We observed mutations in these genes in all wild-type (Table S6) and mutator (Table S9) clones evolved in the complex environments $3A_1$, $3A_2$ and 5A. This efflux pump belongs to the RND (resistance nodulation division) family of efflux pumps, with homologs across many pathogenic species, such as Pseudomonas aeruginosa and Neisseria gonorrhoeae[44,45]. It is a stress-induced efflux pump responsible for exporting bile salts, fatty acids, and heavy metals, as well as antibiotics with diverse mechanisms of action like fluoroquinolones, glycylcyclines, macrolides, β-lactams, and aminoglycosides[44,55]. Our results imply that simultaneous exposure to multiple antibiotics can favour mutations in the genes involved in multi-drug resistance.

Our observations also suggest that such beneficial pleiotropic mutations are rare and most mutations that are retained during evolution are neutral or nearly neutral in our phenotyping environments. The reason is that the number of evolved novel traits neither increases nor decreases systematically with the increasing mutation supply and the amount of retained genetic variation (Fig. S7C, D). Clones evolved in single antibiotic environments further support this point: Even though mutator clones experienced more mutations and retained significantly more genetic variation (Fig. 1B), they did not evolve significantly more novel traits (Fig. 1C).

In addition to having experienced different numbers of mutations, mutator and wild-type clones likely also experienced different kinds of mutations, because mutator strains can produce a spectrum of mutations distinct from those of a wild-type strain[56]. Moreover, the same mutation may produce different fitness effects in the mutator and wild-type strain, even in the early phases of evolution. This is because our mutator ancestor harboured ~25 different mutations compared to the wild-type (Supplementary note S1, Table S2), which may affect the mutator's evolutionary trajectory by interacting epistatically with newly acquired mutations[57]. We emphasise that the accumulation of some mutations in the ancestor during mandatory cultivation steps before evolution, as well as before phenotypic and genomic assays, is one of the unavoidable limitations of any experimental evolution study using a mutator strain[18,51,58]. The differences in mutation rate, spectrum, and genomic background between mutator and wild-type may also affect the rate and extent of novel trait evolution. It is also relevant here that minimum inhibitory concentrations (MICs) for antibiotics might depend on a strain's inherent mutation rate[59–61]. An important direction for future work is to study the role that mutation biases and different ancestral backgrounds might play in the evolution of latent novel traits.

Apart from the different numbers and kinds of mutations, wild-type and mutator strains may have experienced different selection pressures even in identical environments, because the $IC_{90}$ of the five antibiotics for the mutator ancestor was modestly higher or equal to the $IC_{90}$ of the wild-type ancestor (Table S11). This observation prompted us to quantify by how much experimental evolution had increased the $IC_{90}$ for all representative evolved clones. If the differences in the ancestral $IC_{90}$ had resulted in a consistently lower selection pressure on the mutator strain, then mutator strains should have experienced a consistently lower fold-change in the $IC_{90}$ than evolved wild-type clones. However, we observed no such systematic differences (Supplementary fig. S8), suggesting that the ancestral differences in $IC_{90}$ and the fold-changes in $IC_{90}$ after evolution might be poor indicators of this selection pressure. To quantify the selection pressure experienced by populations evolving in complex antibiotic environments at different mutation rates is an important future research direction.

The complex environments in our study exert selection on multiple cellular targets and processes at the same time. Such environments can select for synergistically pleiotropic mutations, i.e., mutations that are beneficial for multiple traits[3,62,63]. Similarly, past work with bacteriophages has shown that synergistically pleiotropic mutations are favoured when phages evolve in a complex environment that selects for both growth rate and capsid stability[62,63]. In contrast, such mutations are not favoured under selection for higher growth rate or increased capsid stability alone. Work like this shows that environments with complex selection pressures can select on pleiotropic mutations to a different extent. Characterising the relationship between environmental complexity and pleiotropy remains another important task for future work. In addition, it will be important to study different kinds of complex environments. For instance, complex environments with more than one carbon source may not favour pleiotropic mutations as strongly as our antibiotic environments do. As a result, fewer latent novel traits may emerge in such environments. Future investigations of this kind will shed light on the generalisability of our results.

By demonstrating the supremacy of selection over mutation at a mutation supply exceeding that of the wild-type, our experiments contribute to a long-standing debate about the forces driving Darwinian evolution. They extend the influence of selection even to traits that are not immediately adaptive. In addition, they have practical implications, especially for the rising incidence of antimicrobial resistance worldwide. Specifically, they caution against the prescription of antibiotic cocktails that can expose a pathogen to multiple antibiotics simultaneously. Not only can such cocktails accelerate the evolution of resistance against antibiotics in the cocktail, they can also endow a pathogen with viability in new and unrelated environments.

## Methods
### Bacterial strains, media and antibiotics
To vary the mutation supply for experimental evolution, we used an *E. coli* strain with a wild-type mutation rate and a mutator strain with a ~22-fold higher mutation rate. We refrained from using an anti-mutator strain with a mutation rate lower than the wild-type, because such strains are rarely encountered in nature[52]. Specifically, we used previously described derivatives of *E. coli* strain K12 MG1655 ($MR^S$ for wild-type, and $MR^L$ for mutator) for experimental evolution[18]. Our mutator strain harbours a 103 bp insertion upstream of the gene *mutL* that is involved in the mismatch repair pathway of *E. coli*. This insertion affects mismatch repair and increases the genomic mutation rate by twenty-two-fold[18].

For all our experiments, we used five different antibiotics, namely trimethoprim, azithromycin, streptomycin, ampicillin and, nalidixic acid (all obtained from Sigma). We chose these antibiotics because their mechanisms of action are diverse, i.e., each targets a different

cellular process[24,25]. We prepared stock solutions of each antibiotic (Table S9) and stored them at −20 °C without any exposure to light. We used LB broth (Sigma) supplemented with the relevant antibiotic for all pilot and evolution experiments.

To prepare a glycerol stock of our ancestral wild-type and mutator strains, we picked a colony for each strain from an LB agar plate, and inoculated it separately in 100 ml LB in two conical flasks without any antibiotic. We incubated each of the flasks at 37 °C with shaking at 220 rpm in a shaking incubator (INFORS HT, Switzerland). After 20 h of growth, we mixed 800 μl samples of bacterial culture with 200 μl of 15% glycerol (v/v) in screw-capped tubes and stored these tubes at −80 °C. We call these the ancestral glycerol stocks of wild-type and mutator strains. We note that the acquisition of some mutations by the mutator population during these obligatory culturing procedures is an inevitable consequence of the mutator's intrinsically high mutation rate.

We determined the $IC_{90}$ for every antibiotic prior to experimental evolution. The $IC_{90}$ is the lowest concentration of antibiotic that is able to reduce a culture's optical density at 600 nm ($OD_{600}$) by 90% compared to the growth of the ancestral strain in the same medium devoid of any antibiotic[1]. As previously described[5] we estimated the $IC_{90}$ for each antibiotic using the following procedure. We inoculated 10 μl of the ancestral wild-type glycerol stock in 3 ml LB and let it to grow for 20 h at 37 °C with shaking at 220 rpm (INFORS HT, Switzerland). We then used 4 μl of this revived culture to inoculate three wells of a 24-well plate (Corning, USA) where each well contained 2 ml of LB supplemented with the antibiotic. We incubated the plate at 37 °C (350 rpm, SI505, Stuart, UK). After 24 h of growth we measured the $OD_{600}$ of the cultures using a plate reader (Tecan, Infinite 200 PRO model using the software Tecan i-control version 3.14). $IC_{90}$ values for all antibiotics are listed in Table S4. Our estimated $IC_{90}$ values are equal to or greater than the clinical breakpoints for *E. coli* suggested by the European Committee on Antimicrobial Susceptibility Testing[64].

After the evolution experiment we determined the $IC_{90}$ values of the five antibiotics for both wild-type and mutator ancestral strains (Table S11). We used the same procedure as just described, with two exceptions: The antibiotics were procured from a different company (Himedia, India, as opposed to Sigma), and the cultures were incubated in an Eppendorf Innova 42 shaker incubator (instead of INFORS HT, Switzerland). We observed a twofold higher $IC_{90}$ of nalidixic acid for the wild-type ancestral strain (Table S11) compared to the $IC_{90}$ determined before the evolution experiment (Table S3). We note that this twofold difference is the smallest measurable MIC difference with our protocol, and suspect that it stems from the change in the antibiotic manufacturer. We considered this two-fold higher value to be the wild-type MIC of nalidixic acid for this set of experiments. We then determined the $IC_{90}$ for all representative clones in the antibiotics experienced during evolution, and used this information to calculate the fold-change in the $IC_{90}$ relative to the corresponding ancestral $IC_{90}$. Multiple clones continued to grow on concentrations of streptomycin and nalidixic acid that were more than 32-fold higher than the wild-type MIC. In the case of nalidixic acid, the antibiotic began to precipitate at the bottom of the culture vessel above a 32-fold increase in concentration. In case of streptomycin, many clones did not decrease their growth at a 32-fold antibiotic concentration increase, which suggests a resistance mechanism independent of the antibiotic concentration. For these reasons, we did not study even higher antibiotic concentrations.

### Experimental evolution for mutator strain in single antibiotic environments
In previously published experiments[5], we had evolved eight populations of a wild-type *E.coli* strain on increasing concentrations of a single antibiotic in five independent experiments (8 populations times 5 antibiotics, i.e., 40 populations in total). We had used the same five antibiotics that we use for the mutator strain in this experiment,

namely ampicillin, azithromycin, nalidixic acid, streptomycin and trimethoprim. At the end of evolution for ~100–200 generations, evolved populations could grow in the $IC_{90}$ of the respective antibiotic. In our current work, we used the same protocol to evolve forty populations of a mutator strain that has a twenty-two fold higher mutation rate than the wild-type strain.

Specifically, we evolved eight replicate populations of the mutator strain on each of the above five antibiotics. We used 24-well plates containing 2 ml LB with antibiotic for experimental evolution, and transferred 4 µl of culture from every evolving population every day. We incubated all the evolving populations at 37 °C with shaking at 350 rpm on an SI505 incubating shaker (SI505, Stuart, UK). We increased the concentration of each antibiotic every 48 h and continued evolution until the populations could grow at the $IC_{90}$ (Table S3). We had performed pilot experiments which had shown that this procedure minimises extinctions and avoids long periods of growth at any one antibiotic concentration. We also chose growth thresholds based on these pilot experiments, which had shown that extinction are common for values of $OD_{600}$ below 0.2, and infrequent for values between 0.2 and 0.3. To avoid extinctions we transferred 20 µl of culture volume instead of 4 µl if a population's $OD_{600}$ (measured on a Tecan Infinite 200 PRO model using the software Tecan i-control version 3.14) had only reached a value between 0.2 and 0.3 after 20 h of incubation. We did this for two populations of mutator strain evolving on azithromycin on the 16th day of experimental evolution. If the growth, measured as $OD_{600}$, of any population was below the threshold of 0.2, we recorded an extinction event for that population. We stored every day's 24-well plates at 4 °C for 72 h. To resume evolution for an extinct population we used 20 µl of inoculum from the same replicate population from the previous day's plate. We did this for one mutator population evolving on azithromycin on day 14.

We checked for contamination by streaking a sample of each population on LB agar plates and visually inspecting these plates after 20 h of incubation at 37 °C. We conducted these purity checks once every week and after the confirmation of purity we stored a part of the population as a glycerol stock at −80 °C. In the event of contamination, we revived the contaminated population from the latest uncontaminated sample by re-inoculating 20 µl of the culture into fresh medium with antibiotic. We observed a single incidence of contamination for populations evolving on single antibiotics. Specifically, we detected that one population evolving on azithromycin was contaminated twice on the second and fourth day of evolution and had to be revived from the first day culture. We note that none of the two representative azithromycin clones we analysed here stem from this population.

We terminated experimental evolution when all the populations were able to grow at the $IC_{90}$ of their respective antibiotic. We then prepared glycerol stocks of all the populations and stored them at −80 °C. The time required to achieve growth at the $IC_{90}$ varied for populations from different antibiotics and was in the range of ~108 to ~215 generations (Table S3). The estimated number of generations is the (base 2) logarithm of the dilution factor we had used for serial transfers[65]. On each antibiotic, mutator and wild-type strains evolved for a similar number of generations until they could grow at the $IC_{90}$[5].

### Experimental evolution in the complex multi-antibiotic environments $3A_1$, $3A_2$, and 5A

We also evolved our wild-type and mutator *E.coli* strains in three different 'complex' environments that contain multiple antibiotics. Two of these environments ($3A_1$ and $3A_2$) contain LB supplemented with three different antibiotics. Specifically, environment $3A_1$ harbours trimethoprim, ampicillin and nalidixic acid. Environment $3A_2$ harbours streptomycin, azithromycin and nalidixic acid ($3A_2$). The third complex environment (5A) harbours all five antibiotics, ampicillin, azithromycin, nalidixic acid, streptomycin and trimethoprim. In each of the $3A_1$, $3A_2$, and 5A environments, we evolved eight replicate populations of the wild-type strain and eight replicate populations of the mutator strain. We used the same initial antibiotic concentrations of each antibiotic as in the single antibiotic environments to initiate experimental evolution in the $3A_1$, $3A_2$ and 5A environments (Table S3). During the first phase of experimental evolution, we increased the concentrations of all antibiotics every second day. We applied the same criteria as during evolution on single antibiotics to determine low growth ($OD_{600}$ between 0.2 and 0.3, 20 µl inoculum volume) and extinction ($OD_{600} < 0.2$, revive 20 µl from the previous day's plate), and to continue the experiment when such an event had taken place.

The second phase was motivated by the observation that population extinctions became more and more frequent while we increased the concentration of all antibiotics simultaneously (Supplementary note S2). For instance, on the seventh day seven out of eight wild-type populations evolving in the 5A environment became extinct. To mitigate this problem, we followed phase I with a phase II in which we modified the evolution protocol in two ways. First we increased the concentration of only one antibiotic every day. For instance, we increased the concentration of streptomycin from 13.9 µg/ml to 14.3 µg/ml on day 112 for the environments $3A_2$ and 5A. This concentration stayed the same for next 4 days, and again increased to 15 µg/ml on day 117. Second, we increased the inoculum volume for serial transfer from 4 µl to 100 µl. We chose this inoculum value based on a pilot experiment that had shown higher rates of extinction for smaller inocula. Subsequently, after all populations had evolved the ability to grow at the $IC_{90}$ of each of the antibiotics in their environment, we decreased the inoculum volume from 100 µl to 50 µl, then to 25 µl, from 25 µl to 12 µl and lastly to 4 µl in 4 days. Once all the evolved populations could grow at the $IC_{90}$ of each of the antibiotics in their environment at a 4 µl inoculum volume we terminated the evolution experiment. We then prepared glycerol stocks of all populations and stored them at −80 °C. Experimental evolution lasted for 147 days or ~600 generations, depending on the strain and antibiotic environment (Supplementary note S2). We estimated this number of generations as the (base 2) logarithm of the dilution factor we had used for serial transfers[65].

Once every week, we streaked a sample of every population on LB agar plates, and inspected the sample visually for contamination after 20 h of incubation at 37 °C. After confirmation of purity, we prepared glycerol stocks and stored them at −80 °C. We detected contamination on day 33 and day 37 in three and one populations, respectively (Supplementary note S2). In every instance of contamination, we examined plated samples of the affected populations from the preceding 3 days, and resumed experimental evolution from the latest uncontaminated sample by re-inoculating 20 µl of culture volume into fresh medium with the appropriate antibiotics.

### Isolation of representative clones

Previously we had isolated two representative clones from eight evolved populations of the wild-type strain for novel trait assays and genome-sequencing[5]. We had selected two ancestral wild-type clones randomly for the same purpose. Using the same protocol, we here chose two representative clones from evolved mutator populations for every one of the five single antibiotic environment, for a total of ten (=2 clones × 5 antibiotics). In addition, we chose two evolved wild-type and two evolved mutator clones from eight populations evolved in each of the complex $3A_1$, $3A_2$ and 5A environments, for a total of 12 clones (=2 strains × 3 environments × 2 clones). Specifically, we streaked a sample of a population's glycerol stock on LB agar and allowed it to grow for 24 h at 37 °C. We randomly selected three colonies from this plate and inoculated them separately in 2 ml LB without any antibiotic. We let the liquid cultures grow for 20 h at 37 °C in a shaking incubator

at 220 rpm (INFORS HT, Switzerland) and then stored them as glycerol stocks at −80 °C.

We revived 4 μl of glycerol stock in 200 μl of LB in a 96-well plate (Thermo) for all the isolated clones, as well as for the eight replicate populations evolved in every environment. We incubated the plate overnight at 37 °C in a shaking incubator (350 rpm, SI505, Stuart, UK). We inoculated 4 μl of revived culture into 200 μl of medium with the antibiotic environment that the clone or population had experienced on the last day of experimental evolution, i.e., with either one, three, or five antibiotics at their respective $IC_{90}$. For the next 24 h we tracked the growth of all the clones and populations by measuring their $OD_{600}$ every 15 min using a plate reader (Tecan Infinite 200PRO model using the software Tecan i-control version 3.14). This assay gave us eight different whole-population growth trajectories, and twenty-four growth trajectories for the clones isolated from the eight populations, for every combination of an environment and a strain. We then used the Growth Rates[66] software to determine the growth rates from each of the growth trajectories. We computed 95% confidence intervals for the mean growth rate from the eight whole-population growth trajectories for a given antibiotic environment × strain combination, and identified those clones whose growth rate lay within these 95% confidence intervals (Fig. S4). The rationale behind this choice is that these clones are the best representatives of the central tendency of the populations that evolved in the antibiotic(s). From this set of identified clones, we randomly chose two clones for each environment × strain combination for further experiments. Including more representative clones in our phenotypic and genomic analysis was prohibitive due to financial and logistic limitations. We emphasise that this sample size of two for every combination of strain and environment is a limitation of our study.

In addition to the representative clones we also chose two ancestral clones for further analysis. For this purpose we plated the ancestral mutator strain on an LB agar plate and incubated the plate overnight at 37 °C. We randomly chose two clones from this plate and inoculated all the twenty-two evolved and two mutator ancestral clones, 24 in total, in 2 ml LB and allowed them to grow for 20 h at 37 °C at 220 rpm (Table S5, Fig. S4). We stored a part of these overnight cultures as glycerol stocks and used them for phenotypic assays and genomic DNA extraction.

## Novel trait assays

As also described previously for wild-type clones evolved in single antibiotic environments[5], we used ten phenotyping microarrays (Biolog PM11-20, Biolog, CA, US)[67] to study the evolution of latent novel traits in the evolved clones. Biolog microarrays are 96-well plates containing preconfigured sets of antimicrobials along with a tetrazolium dye as an indicator for cell respiration. During respiration bacterial cells produce NADH, which reduces the tetrazolium dye to produce a purple colour. The intensity of the purple colour can be easily measured using a spectrophotometer and correlates with the magnitude of respiration. The set of ten Biolog plates contain 236 potentially bactericidal or bacteriostatic molecules at four different concentrations (240 molecules as per the manufacturer, but see ref. 69). The actual concentration range of each molecule varies among molecules, and is proprietary information of the manufacturer. Antimicrobial substances in the Biolog arrays include, but are not limited to, antibiotics, organic and inorganic salts, nucleotide analogues, pyridine derivatives, and surfactants. We used DrugBank, PubChem, and original research articles[68,69] to collect information on the mode of action of these antimicrobials. We could not find any relevant information for 38 out of the 236 molecules.

We next determined the environments which support the growth of ancestral and/or evolved clones. To this end, we revived 4 μl of glycerol stocks (~3 × 10⁴ cells) of every evolved and ancestral clone in

5 ml NB and allowed it to grow overnight at 37 °C in an incubating shaker (220 rpm, INFORS HT, 417 Switzerland). We prepared an inoculating mix by adding ~2 ml of this overnight grown culture culture (~2 × 108 cells) to 100 ml of inoculating fluid (IF-10 reagent by Biolog, US). Subsequently, we added 100 μl (~2 × 105 cells) of this inoculating mix to every phenotyping environment (each well in the set of ten 96-well plates, PM11-20, contains a unique phenotyping environment). We incubated the Biolog plates at 37 °C for 48 h (SI505, Stuart, UK). We measured the $OD_{600}$ immediately after inoculation (0 h) and after 48 h of growth using a plate reader (Tecan, Infinite 200 PRO model using the software Tecan i-control version 3.14). We accounted for the instrument and inoculum background by subtracting the $OD_{600}$ measurement at 0 h from the $OD_{600}$ measured at 48 h. If both ancestral clones showed an $OD_{600}$ below 0.3 after 48 h, while both evolved clones from the same antibiotic showed an $OD_{600}$ above 0.3 after 48 h, we considered a novel trait to have evolved. We used this growth threshold based on the previous finding that inoculating 230 of the Biolog environments yields an $OD_{600}$ below 0.3 immediately after inoculation, i.e., at zero hours[5].

We considered only the highest antimicrobial concentration for each of the 236 phenotyping environments, because the wild-type ancestor was viable on the three lower concentration of most antimicrobials in the phenotyping environemnts[5]. As a result, the lower three concentrations presented very little opportunity for the evolution of novel traits. We found that both ancestral clones were unable to grow at the highest antimicrobial concentration for 95 environments[5] while both mutator ancestral clones could not grow at the highest concentration for 58 environments. Thus, the mutator strain could acquire viability in at most 58 environments. For the determination of novel trait evolution, we considered only those 52 environments where neither wild-type nor mutator ancestor was viable (Table S1).

## Whole-genome sequencing of ancestral and evolved clones

As we had done previously for wild-type clones, we extracted genomic DNA for genome sequencing from two ancestral mutator clones, and from two mutator clones that had evolved in each of the single antibiotic environments. In addition, we extracted the DNA of two wild-type and two mutator clones evolved in environments $3A_1$, $3A_2$, and 5A. We performed DNA extractions using the DNeasy Blood and tissue kit from Qiagen (catalogue no 69504). Specifically, we inoculated 4 μl of glycerol stock of each clone in 5 ml of LB without antibiotic. We let this culture grow for ~16 h at 37 °C with shaking at 220 rpm (INFORS HT, Switzerland). We centrifuged the cultures at 10,000 $g$ (Eppendorf 5810/5810R) for 10 min to harvest ~2 × 10⁹ cells. We used the kit's protocol to extract the DNA from the harvested cells. Using a Qubit fluorometer (Thermo Fisher Scientific) and agarose gel electrophoresis we checked the quantity and purity of the extracted DNA and stored this DNA at −20 °C. The whole genome of each clone was sequenced using the Illumina HiSeq (Illumina, CA, USA) at MicrobesNG (Oxford, UK) to a minimum coverage of 30-fold per clone. MicrobesNG provided us with the trimmed reads as fastq files. We identified mutations in these sequences using the Breseq pipeline v0.35 with default parameters[70]. We characterised novel mutations as those mutations that were not present in the wild-type ancestor[5]. We compared the obtained reads with reads from the wild-type ancestor at the locus of mutation using the Integrative Genomics Viewer (v2.9.2, Broad Institute, CA, US) and visually confirmed each mutation identified by Breseq. We annotated the function of each mutated gene using curated descriptions on EcoCyc and references therein[35].

## Statistical analysis

To compare the number of mutations in the wild-type and mutator clones evolved in single antibiotic environments (Fig. 1B), we used an analysis of variance with the two fixed factors of strain (wild-type or mutator) and antibiotic (ampicillin or azithromycin or nalidixic acid or

streptomycin or trimethoprim). To compare the percentage of novel traits between wild-type and mutator clones evolved in single-antibiotic environments (Fig. 1C) we used Wilcoxon rank sum tests (also called Mann–Whitney tests).

We used Spearman's rank correlation coefficient to quantify the association between the percentage of novel traits and the number of antibiotics in the evolution environment both for the wild-type (Fig. 2B) and the mutator strain (Fig. 2C). We performed again an analysis of variance with two fixed factors, i.e., strain (wild-type or mutator) and environment ($3A_1$ or $3A_2$ or 5A) to compare the percentage of novel traits in the wild-type and mutator clones evolved in complex antibiotic environments.

We performed individual Wilcoxon rank sum tests to compare the total number of genomic mutations (Fig. 3A) and the percentage of novel traits (Fig. 3B) between the wild-type clones evolved in complex environments and mutator clones evolved in simple environments. We performed a partial correlation analysis to control for the confounding effect of the duration of experimental evolution in the association between the percentage of novel traits and the complexity of the environment. We quantified the association using Spearman's rank correlation coefficient for both the wild-type and the mutator strains.

We used R (v3.5.2) for all statistical analyses.

## Reporting summary

Further information on research design is available in the Nature Research Reporting Summary linked to this article.

## Data availability

All data are available in the paper, the supplementary materials and source data file. Whole genome sequencing data of the bacterial isolates is available from NCBI with Bioproject number PRJNA882999. Source data are provided with this paper.

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

## Acknowledgements

This project has received funding from the European Research Council under Grant Agreement No. 739874. We would also like to acknowledge support by Swiss National Science Foundation grant 31003A_172887 and by the University Priority Research Program in Evolutionary Biology. We thank Dr. Sutirth Dey for providing lab infrastructure and consumables for follow-up experiments performed during manuscript revision, as well as for valuable discussions.

## Author contributions

S.K. and A.W. were involved in the conceptualisation of the study. S.K. designed and performed the experiments and wrote the paper. S.K. and A.W. contributed to data analysis and edited the paper.

## Competing interests

The authors declare no competing interests.
