## [Peer Review File · Nature Communications]

Environmental complexity is more important than mutation in driving the evolution of latent novel traits in *E. coli*Editorial Note: This manuscript has been previously reviewed at another journal that is not operating a transparent peer review scheme. This document only contains reviewer comments and rebuttal letters for versions considered at Nature Communications. Mentions of prior referee reports have been redacted.

Reviewers' Comments:

Reviewer #1:

Remarks to the Author:

Overall, the revised manuscript is clearer and deals with caveats and nuances much better, and I applaud the authors for this. Unfortunately, I remain concerned about some key points: the authors come to very broad and general conclusions in the manuscript (mutation supply is not important, but environmental complexity is important for the evolution of novel traits), and such claims need commensurate support.

First, I do not agree with the assertion (in the reviewer response) that comparing any two mutation rates is informative to test the role of mutation supply. Suppose the evolution of novelty is an increasing saturating function of (relevant) genetic variation. If you are comparing two points towards the very right of this plot, you will come to very different conclusions (no effect of mutation supply or genetic variation) than if you compared two points at extreme ends of the curve (large increase in novelty as a function of mut supply/gen var). So, to make a general conclusion about mutation supply one would need to have sampled at least one point representing a substantially lower mutation supply. In their response, the authors say that they chose not to use anti-mutators because they are uncommon in nature, but the cited papers actually do not directly test or show this. Further, there are many species that have much lower mutation rates than *E. coli*, so this argument is weak.

Second, as noted by all reviewers, there are many differences between the wild type and mutator (background mutations, mutation spectrum, initial fitness, clonal interference), in addition to mutation supply. Hence, to make general conclusions about mutation supply one would either need to test a diverse set of mutators or remove the effect of these other confounding factors. This is important because these factors may affect the fate of new mutations (hence confounding the impact of supply), but not the impact of environmental complexity directly. When confounding factors affect only one of two variables of interest, it is difficult to know whether one of them truly has no impact or these other confounding factors mask those true effects. E.g. clonal interference may prevent the effect of mutation supply from being apparent, which is not the same as saying that mutation supply is not important.

Finally, the sample size of two replicates remains an issue and reduces the ability to make general conclusions.

I do understand that logistical challenges prevent extensive experiments; but then I would like to see an appropriate toning down of the generality of the presented conclusions.

Reviewer #2:

Remarks to the Author:

The authors have revised and significantly improved the manuscript by applying the suggested new analysis.

One of my important comments concerns the evolution experiment: I am afraid that the two different

strains (wild type/mutator) may have experienced different selection pressure during the evolution experiment simply because of the different initial MIC of the two strains. What were the final MIC fold changes (compared to the initial MIC of the strains) of the evolved wild type and mutator lines? If the selection pressure was substantially different on the two strains, would it affect their general conclusions?

How many of the "novel traits" actually involve antimicrobials from different antibiotic classes than the "evolution environment" in Figure 1C and Figure 2B,C? I would not consider for example a trait "novel" if the strains acquired viability to an aminoglycoside following the evolution to another type of aminoglycoside. Their definition of novel traits should be more constrained and only consider a trait novel if the evolved lines acquired viability to a completely new drug from different classes or detergents.. etc.

Reviewer #3:

Remarks to the Author:

[Redacted]

In this revised version, the authors have addressed this only concern I had (regarding the initial genotypic and phenotypic differences between the wildtype and mutator ancestral strains). Their analysis of the 52 environments in which neither ancestral strain is able to grow supports their claims, and thus completely dissipates my concern.

I thus fully recommend publication.

Antoine Frenoy

Reviewer #1:

Reviewer #1: Overall, the revised manuscript is clearer and deals with caveats and nuances much better, and I applaud the authors for this. Unfortunately, I remain concerned about some key points: the authors come to very broad and general conclusions in the manuscript (mutation supply is not important, but environmental complexity is important for the evolution of novel traits), and such claims need commensurate support.

Response: Thank you very much for appreciating our efforts! We completely agree with you that the conclusion in the previous draft needed more nuance. We have now toned down the narrative and clarified the limitations of our results. Below are all the instances where we have modified the narrative:

Abstract

Lines 26-28 "*Our experiments show that the selection pressure provided by an environment can be more important for the evolution of novel traits than the mutational supply experienced by a wild-type and a mutator strain of E. coli.*"

Introduction

Lines 82-84 "*We found that for the wild-type and mutator strains the number of latent novel traits that evolved depended on the complexity of the environment but not on the mutation supply.*"

Lines 86-87 "*In sum, at the mutation rates of our wild-type and mutator strains, selection exerted by the environment is the key force in the evolution of novel traits without immediate benefits.*"

Results

Lines 168-169 "*Thus, the supply of mutations does not limit the evolution of latent novel traits at mutation rates that exceed those of the wild-type.*"

Discussion

Lines 342-347 "*To study how readily latent novel traits may evolve at mutation rates lower than that of the wild-type remains an exciting direction for future work. Additionally, one could also replicate our experimental design in a bacterial species with a lower wild-type mutation rate than that of E. coli. Results of such studies, combined with our observations, can uncover the relationship between mutation rates and the emergence of latent novel traits over a wider range of mutation supplies.*"

Lines 418-420 "*By demonstrating the supremacy of selection over mutation at a mutation supply exceeding that of the wild-type, our experiments contribute to a long-standing debate about the forces driving Darwinian evolution.*"

Reviewer #1: First, I do not agree with the assertion (in the reviewer response) that comparing any two mutation rates is informative to test the role of mutation supply. Suppose the evolution of novelty is an increasing saturating function of (relevant) genetic variation. If you are comparing two points towards the very right of this plot, you will come to very different conclusions (no effect of mutation supply or genetic variation) than if you compared two points at extreme ends of the curve (large increase in novelty as a function of mut supply/gen var). So, to make a general conclusion about mutation supply one would need to have sampled at least one point representing a substantially lower mutation supply. In their response, the authors say that they chose not to use anti-mutators because they are uncommon in nature, but the cited papers actually do not directly test or show this. Further, there are many species that have much lower mutation rates than *E. coli*, so this argument is weak.

Response: Thank you very much for raising this concern. We stand corrected, and agree with you that we do not know the nature of the relationship between mutation supply and the emergence of novelty beyond the two mutation rates we study. If novelty is a saturating function of the mutation supply whose saturation occurs above wild-type mutation rates, as you hypothesize, then the relationship our paper has identified may hold only for mutation rates above the wild-type. However, latent novel traits remain poorly studied and we currently have no indication of the nature of the function that relates the incidence of latent novel traits to mutation supply. Our work provides at least two "data points" for this relationship.

In sum, you have identified an important limitation of our analysis, and we now state this limitation in multiple places. (Please see our responses to your previous concern above.) For example, we now identify this limitation in the discussion and suggest that an analysis of the relationship between mutation supply and the emergence of latent traits is an obvious important direction for future work.

Lines 342-343 "*To study how readily latent novel traits may evolve at mutation rates lower than that of the wild-type remains an exciting direction for future work.*"

We also agree with you that mutation rates of many bacterial species are lower than *E. coli*. We would like to point out, however, that comparing the extent of latent trait evolution across different species may be misleading due to many differences in the ancestral genotypes and phenotypes of the two species. A more meaningful comparison might involve strains with two different mutation rates that belong to a species which has a different wild-type mutation rate than *E. coli*. We now include such a study as another important future direction.

Lines 344-347 "*Additionally, one could also replicate our experimental design in a bacterial species with a lower wild-type mutation rate than that of E. coli. Results of such studies, combined with our observations, can uncover the relationship between mutation rates and the emergence of latent novel traits over a wider range of mutation supplies.*"

Reviewer #1: Second, as noted by all reviewers, there are many differences between the wild type and mutator (background mutations, mutation spectrum, initial fitness, clonal interference), in addition to mutation supply. Hence, to make general conclusions about mutation supply one would either need to test a diverse set of mutators or remove the effect of these other confounding factors. This is important because these factors may affect the fate of new mutations (hence confounding the impact of supply), but not the impact of environmental complexity directly. When confounding factors affect only one of two variables of interest, it is difficult to know whether one of them truly has no impact or these other confounding factors mask those true effects. E.g. clonal interference may prevent the effect of mutation supply from being apparent, which is not the same as saying that mutation supply is not important.

Response: We agree with you that the wild-type and mutator strains have many differences. We have limited the contribution of these differences by restricting our analysis to only those 52 environments where neither wild-type nor mutator ancestor could grow. Despite this conservative analysis, differences between the genotypes of wild-type and mutator ancestors can interact with the newly arising mutations in different ways and affect the fate of those mutations. This is another limitation of our work that you have correctly identified, and that we now discuss in detail:

Lines 398-408 "*Apart from the different numbers and kinds of mutations, wild-type and mutator strains may have experienced different selection pressures even in identical environments, because the IC_{90} of the five antibiotics for the mutator ancestor was modestly higher or equal to the IC_{90} of the wild-type ancestor (Table S19). This observation prompted us to quantify by how much experimental evolution had increased the IC_{90} for all representative evolved clones. If the differences in the ancestral IC_{90} had resulted in a consistently lower selection pressure on the mutator strain, then mutator strains should have experienced a consistently lower fold-change in the IC_{90} than evolved wild-type clones. However, we observed no such systematic differences (Supplementary figure S20), suggesting that the ancestral differences in IC_{90} and the fold-changes in IC_{90} after evolution might be poor indicators of this selection pressure. To quantify the selection pressure experienced by populations evolving in complex antibiotic environments at different mutation rates is an important future research direction.*"

Reviewer #1: Finally, the sample size of two replicates remains an issue and reduces the ability to make general conclusions. I do understand that logistical challenges prevent extensive experiments; but then I would like to see an appropriate toning down of the generality of the presented conclusions.

Response: Thank you very much for pointing this out. In response to this and the first comment we have now revised the manuscript extensively and identified the limitations of our conclusion clearly. In addition to the changes listed in response to the comments, we have now included the following lines in the current draft of the manuscript.

Lines 590-592 "*Including more representative clones in our phenotypic and genomic analysis was prohibitive due to financial and logistic limitations. We emphasize that this sample size of two for every combination of strain and environment is a limitation of our study.*"

Reviewer #2:

Reviewer #2: The authors have revised and significantly improved the manuscript by applying the suggested new analysis.

Response: Thank you very much! Below we answer your two remaining concerns in a detailed manner.

Reviewer #2: One of my important comments concerns the evolution experiment: I am afraid that the two different strains (wild type/mutator) may have experienced different selection pressure during the evolution experiment simply because of the different initial MIC of the two strains. What were the final MIC fold changes (compared to the initial MIC of the strains) of the evolved wild type and mutator lines? If the selection pressure was substantially different on the two strains, would it affect their general conclusions?

Response: Thank you very much for raising this concern. We agree with you that the strength of selection experienced by the wild-type and mutator populations might have been different. To find out, we now conducted new experiments in which we measured the fold-changes in the MIC (measured as IC_{90} in $\mu\text{g/ml}$, methods) for all representative clones in all antibiotics they experienced during the experimental evolution.

First, we determined the MIC of the five antibiotics for the mutator ancestral strain, using the same protocol for MIC determination as for our main analysis (lines 441-451 in the main text). Table R1 below (now also included in the supplementary text as Table S19) lists the MICs of the five antibiotics for the wild-type and the mutator ancestral strain. For the interpretation of MICs, it is important that we follow most other authors by measuring them on a logarithmic scale, by a successive doubling of antibiotic concentration (please see *Clinical Microbiology and Infection* 6.9

(2000): 509-515 by EUCAST and ESCMID; Nature protocols 3.2 (2008): 163-175). Thus, the smallest measurable change in MIC is a doubling of an antibiotic's concentration.

Ancestral strain	Ampicillin	Azithromycin	Nalidixic acid	Streptomycin	Trimethoprim
Wild-type	8	25.6	32	16	409.6
Mutator	8	51.2	128	32	819.2

Table R1: MIC (measured as IC_{90} in $\mu\text{g/ml}$, methods) of the five antibiotics for the wild-type and mutator ancestral strains.

Table R1 shows that the mutator ancestor had a two-fold higher MIC for azithromycin, streptomycin and trimethoprim than the wild-type ancestor. The MIC for nalidixic acid was four-fold higher and that of ampicillin was the same as in the wild-type ancestor. Thus in general, the MICs for the mutator ancestor were only slightly higher or equal to the wild-type ancestor. Our results are compatible with previous observations that MICs of mutators may or may not be higher than the wild-type (for some examples of such differences between the MICs of wild-type and mutator see Table 1 from Ellington et al. (2006) *Journal of Antimicrobial Chemotherapy*, 58 (4): 848–852, Figure 3 from Mehta et al. (2019) *Antimicrobial Agents Chemotherapy*. 63 (7):e00744-19. and Watson et al. (2004) *Microbiology* 150 (9): 2947-2958).

Does this modest difference in the ancestral MICs translate into different fold-changes in MIC after evolution? To find out, we measured the fold-changes of our representative evolved clones in the MIC (Figure R1 below, new Supplementary Figure S20) for the antibiotics experienced during evolution, i.e., one antibiotic for the clones from single antibiotic environments (Figure R1A), three antibiotics for clones from the 3A₁ and 3A₂ environments (Figure R1B and R1C), and five antibiotics for the clones from the 5A environment (Figure R1D). In total, we thus measured MICs for 32 (=2x2x8) representative clones in total, two clones for every combination of the mutation rate and evolution environment. We again used the same protocol for MIC determination as in our main analysis (lines 441-451 in the main text). We continued increasing the concentration of each antibiotic until the concentration was 32-fold higher than the MIC of the wild-type ancestor (please refer to methods section for details). The clones that continued to grow at this concentration are marked with a downward arrow ('↓') in Figure R1. We calculated the fold-change in MIC relative to the corresponding ancestral strain.

The observation (Table R1) that the mutator ancestor showed slightly higher MICs than the wild-type ancestor raised the possibility of a systematically weaker selection pressure experienced by the mutator clones. If so, the fold-changes in the MIC for our evolved mutator clones should be lower, or at least equal, but not higher than those of the evolved wild-type clones. However, this

is not generally the case, at least not systematically: Some mutator clones showed greater fold-changes than the wild-type despite supposedly weaker selection pressure. A case in point are the mutator clones evolved in the 3A₂ environment (Figure R1C). The MIC of azithromycin for these mutator clones increased more than 32-fold, while that of the wild-type clones increased only modestly (two-fold and eight-fold each). In other cases, the MIC changed more in wild-type clones than in mutator clones, e.g., for trimethoprim in the 3A₁ environment. And the MIC changed dramatically for both mutator and wild-type clones on nalidixic acid, regardless of the evolution environment (Figure R1). Another notable case is the MIC for ampicillin, which changed to a greater extent in the mutator strain than in the wild-type (Figure R1A, R1C and R1D) even though both ancestors had the same MIC.

Figure S20: Fold-change in MIC (measured as IC₉₀ in µg/ml, methods) for 32 representative clones on the antibiotics experienced during experimental evolution. A. Fold-changes in MIC for the two wild-type and two mutator clones evolved in five different single antibiotic environments. **B.** Fold-changes in MIC for the two wild-type and two mutator clones evolved in the 3A₁ environment that contained ampicillin, nalidixic acid and trimethoprim. **C.** Fold-changes in MIC for the two wild-type and two mutator clones evolved in the 3A₂ environment that contained azithromycin, nalidixic acid and streptomycin. **D.** Fold-changes in MIC for the two wild-type and two mutator clones evolved in the 5A environment, which contained all the five antibiotics. For every clone, the fold-change in MIC is shown relative to the MIC of the

corresponding ancestral strain. Vertical downward-facing arrows '↓' indicate changes greater than 32-fold compared to the wild-type ancestor.

Overall these results suggest that ancestral differences in MIC and fold-changes in the MIC after evolution are poor indicators of the selection pressure experienced during the evolution. How to quantify the selection pressure experienced by populations of different mutation rates evolving in complex antibiotic environments is an important outstanding question for future work, but remains out of the scope of this work. We now include these experiments and their findings in the manuscript, and discuss them in the following places.

Methods

Lines 462:479 *"After the evolution experiment we determined the IC_{90} values of the five antibiotics for both wild-type and mutator ancestral strains (Table S19). We used the same procedure as just described, with two exceptions: The antibiotics were procured from a different company (Himedia, India, as opposed to Sigma), and the cultures were incubated in an Eppendorf Innova 42 shaker incubator (instead of INFORS HT, Switzerland). We observed a two-fold higher IC_{90} of nalidixic acid for the wild-type ancestral strain (Table S19) compared to the IC_{90} determined before the evolution experiment (Supplementary text, Table S4). We note that this two-fold difference is the smallest measurable MIC difference with our protocol, and suspect that it stems from the change in the antibiotic manufacturer. We considered this two-fold higher value to be the wild-type MIC of nalidixic acid for this set of experiments. We then determined the IC_{90} for all representative clones in the antibiotics experienced during evolution, and used this information to calculate the fold-change in the IC_{90} relative to the corresponding ancestral IC_{90} . Multiple clones continued to grow on concentrations of streptomycin and nalidixic acid that were more than 32-fold higher than the wild-type MIC. In the case of nalidixic acid, the antibiotic began to precipitate at the bottom of the culture vessel above a 32-fold increase in concentration. In case of streptomycin, many clones did not decrease their growth at a 32-fold antibiotic concentration increase, which suggests a resistance mechanism independent of the antibiotic concentration. For these reasons, we did not study even higher antibiotic concentrations."*

Discussion

Lines 498:408 *"Apart from the different numbers and kinds of mutations, wild-type and mutator strains may have experienced different selection pressures even in identical environments, because the IC_{90} of the five antibiotics for the mutator ancestor was modestly higher or equal to the IC_{90}*

of the wild-type ancestor (Table S19). This observation prompted us to quantify by how much experimental evolution had increased the IC_{90} for all representative evolved clones. If the differences in the ancestral IC_{90} had resulted in a consistently lower selection pressure on the mutator strain, then mutator strains should have experienced a consistently lower fold-change in the IC_{90} than evolved wild-type clones. However, we observed no such systematic differences (Supplementary figure S20), suggesting that the ancestral differences in IC_{90} and the fold-changes in IC_{90} after evolution might be poor indicators of this selection pressure. To quantify the selection pressure experienced by populations evolving in complex antibiotic environments at different mutation rates is an important future research direction."

Reviewer #2: How many of the "novel traits" actually involve antimicrobials from different antibiotic classes than the "evolution environment" in Figure 1C and Figure 2B,C? I would not consider for example a trait "novel" if the strains acquired viability to an aminoglycoside following the evolution to another type of aminoglycoside. Their definition of novel traits should be more constrained and only consider a trait novel if the evolved lines acquired viability to a completely new drug from different classes or detergents.. etc.

Response: Thank you for raising this issue. We completely agree with you that calling a trait novel may not be appropriate if all antimicrobials from the phenotyping environments in which a clone had evolved viability share their mechanism of action with the antimicrobial from the evolution environment. In our previous work we had demonstrated that many novel traits involve antimicrobials from a drug-class other than that of the evolution environment (Figure 3B from Karve and Wagner (2021), *Molecular Biology and Evolution* 39 (1):msab341). In response to this concern, we performed a similar analysis for the novel traits reported in this work (supplementary table 18). Briefly, out of 22 novel traits (12 in the wild-type and 10 in the mutator strain) that evolved in simple environments (Figure 1C in main text), 15 (8 in the wild-type and 7 in the mutator strain) novel traits involved antimicrobials with a mechanism of action different from that of the antimicrobial in the evolution environment. Similarly, out of 36 novel traits (21 in the wild-type and 15 in the mutator strain) that evolved in the complex environments, 13 novel traits (8 in the wild-type and 5 in the mutator strain) novel traits involved antimicrobials with a mechanism of action different from that of *all* the antimicrobials in the respective evolution environment.

This analysis clearly shows that novel viability evolves on antimicrobials from diverse drug-classes, and is not restricted to those antimicrobials which share their mechanism of action with the antimicrobial from the evolution environment.

Reviewer #3:

Reviewer #3:

[redacted]

In this revised version, the authors have addressed this only concern I had (regarding the initial genotypic and phenotypic differences between the wildtype and mutator ancestral strains). Their analysis of the 52 environments in which neither ancestral strain is able to grow supports their claims, and thus completely dissipates my concern.

I thus fully recommend publication.

Antoine Frenoy

Response: Thank you very much for your constructive suggestions on the previous draft of the manuscript. It helped us make the narrative clearer.

Reviewers' Comments:

Reviewer #1:

Remarks to the Author:

I am satisfied with the latest revisions.

Reviewer #2:

Remarks to the Author:

In this revised version, the authors have addressed my last main concern about the possible differences in the selection pressure experienced by the wild-type and mutator populations during the evolution experiment. Overall, this revised manuscript contains important additional experimentally measured data that have further strengthened their conclusion.

I have only one last comment. Another limitation of their study is that the simple and complex evolution environments only involved antibiotics. However, antibiotics are known to generally enhance the mutation rates of the target cells, thereby accelerating the rate of adaptation even for the wild type strain by this. Therefore, it would be important to mention in the discussion that future studies investigating the effect of complex environments and mutational supply on the evolution of novel traits in other selection environments than antibiotics would be also important to uncover the generalizability of the findings.

Reviewer #1: I am satisfied with the latest revisions.

Response: Thank you very much!

Reviewer #2: In this revised version, the authors have addressed my last main concern about the possible differences in the selection pressure experienced by the wild-type and mutator populations during the evolution experiment. Overall, this revised manuscript contains important additional experimentally measured data that have further strengthened their conclusion.

Response: We thank you for your valuable comments and criticism. They helped us improve the quality of the manuscript substantially.

Reviewer #2: I have only one last comment. Another limitation of their study is that the simple and complex evolution environments only involved antibiotics. However, antibiotics are known to generally enhance the mutation rates of the target cells, thereby accelerating the rate of adaptation even for the wild type strain by this. Therefore, it would be important to mention in the discussion that future studies investigating the effect of complex environments and mutational supply on the evolution of novel traits in other selection environments than antibiotics would be also important to uncover the generalizability of the findings

Response: We completely agree with you. Evolution in non-antimicrobial complex environments may not result in the evolution of latent traits to an extent that we observed in our experiments. The generalizability of our findings thus needs to be checked by future studies. We now clarify this limitation in the discussion.

Lines 383-388 "*Characterizing the relationship between environmental complexity and pleiotropy remains another important task for future work. In addition, it will be important to study different kinds of complex environments. For instance, complex environments with more than one carbon source may not favor pleiotropic mutations as strongly as our antibiotic environments do. As a result, fewer latent novel traits may emerge in such environments. Future investigations of this kind will shed light on the generalizability of our results.*"